# Oxygen Induced Phase Transformation in TC21 Alloy with a Lamellar Microstructure

**Shu Wang** [1,2,3]**, Yilong Liang** [1,2,3,]*****, Hao Sun** [1,2,3]**, Xin Feng** [1,2,3] **and Chaowen Huang** [1,2,3]

1   College of Materials and Metallurgy, Guizhou University, Guiyang 550025, China;
    18946678927@163.com (S.W.); sunhao20201225@163.com (H.S.); fx951001@163.com (X.F.);
    cwhuang@gzu.edu.cn (C.H.)
2   Key Laboratory for Mechanical Behavior and Microstructure of Materials of Guizhou Province,
    Guiyang 550025, China
3   National & Local Joint Engineering Laboratory for High-Performance Metal Structure Material and
    Advanced Manufacturing Technology, Guiyang 550025, China
*   Correspondence: ylliang@gzu.edu.cn

**Abstract:** The main objective of the present study was to understand the oxygen ingress in titanium alloys at high temperatures. Investigations reveal that the oxygen diffusion layer (ODL) caused by oxygen ingress significantly affects the mechanical properties of titanium alloys. In the present study, the high-temperature oxygen ingress behavior of TC21 alloy with a lamellar microstructure was investigated. Microstructural characterizations were analyzed through optical microscopy (OM), scanning electron microscopy (SEM), electron backscatter diffraction (EBSD), and transmission electron microscopy (TEM). Obtained results demonstrate that oxygen-induced phase transformation not only enhances the precipitation of secondary $\alpha$-phase ($\alpha$s) and forms more primary $\alpha$ phase ($\alpha$p), but also promotes the recrystallization of the ODL. It was found that as the temperature of oxygen uptake increases, the thickness of the ODL initially increases and then decreases. The maximum depth of the ODL was obtained for the oxygen uptake temperature of 960 °C. In addition, a gradient microstructure ($\alpha$p + $\beta$ + $\beta$trans)/($\alpha$p + $\beta$trans)/($\alpha$p + $\beta$) was observed in the experiment. Meanwhile, it was also found that the hardness and dislocation density in the ODL is higher than that that of the matrix.

**Keywords:** titanium alloy; oxygen diffusion; phase transformation; gradient microstructure; recrystallization

## 1. Introduction

Studies reveal that oxygen is one of the most important interstitial solutes in titanium alloys, which has a significant impact on the microstructure and mechanical properties of Ti alloys. In hot working processes, such as hot rolling and hot forging, the oxygen diffusion layer (ODL), as an addition to the compound layer is formed in the subsurface of the wrought titanium. For instance, when the heat-treatment temperature or thermal-mechanical treatment temperature of titanium alloys exceeds 480 °C, the alloy surface rapidly oxidizes and a thin $\alpha$-case layer forms on the surface [1]. Further investigations reveal that the existence of oxygen in titanium alloys significantly decreases stress-corrosion, cracking resistance, and fracture toughness in Ti alloys [2]. In addition, oxygen can also provide solid solution strengthening with almost no impact on the corrosion resistance [3]. However, recent investigations demonstrated that the disadvantages induced by the oxygen ingress in titanium alloys are much greater than its advantages. In particular, the presence of an $\alpha$-case layer reduces some important mechanical properties of a titanium alloy, including ductility, fracture toughness, and fatigue life of titanium alloys [4–6]. Therefore, removal of the surface layer is indispensable in manufacturing hot-worked titanium products. It is worth noting that different methods, including grinding, polishing, and chemical milling can be applied to remove the oxidized surface layer. However, these processes remarkably

increase the manufacturing cost of titanium workpieces in comparison with other structural materials such as steel and aluminum alloys. Accordingly, investigating the oxidation characteristics of titanium alloys has become a research hot spot in recent years [7].

TC21 alloy is a relatively new member of the Ti–Al–Sn–Zr–Mo–Cr–Nb–Si series of dual-phase titanium alloys [8]. This alloy has superior characteristics such as low density, high strength, high fracture toughness, and reasonable corrosion resistance. Accordingly, this alloy has been widely applied in structural forgings and bearing parts in diverse industries, including aerospace, marine applications, chemistry, power plants, and automobile [9–12]. Currently, studies on TC21 alloy are mainly focused on its mechanical properties and microstructure evolution [13,14]. However, there are few investigations about the oxygen ingress impact on titanium alloys in α + β and β fields. Exposure of titanium alloys to the high-temperature oxygen-containing atmosphere leads to the formation of ODLs. On the other hand, the generated ODL is hard and brittle, thereby degrading the alloy plasticity. Moreover, the formation and thickness of ODLs affect the life span of the specimen made of titanium alloys [15]. TC21 alloy is mainly applied to manufacture structural members of wing joints and engine joints in the aerospace industry. When the engine runs, the temperature inside the engine exceeds 1000 °C. Meanwhile, the friction between wings and the surrounding air dramatically increases the wing surface temperature during the flight. In both conditions, the oxygen in the surrounding environment easily diffuses into the material, thereby reducing the lifetime of the engine and increasing the possibility of the wing fracturing.

As mentioned earlier, although some researchers have reported the effect of temperatures on the oxidation behavior of titanium alloys, these studies focused on the thickness of the compound layer, the effect of temperatures on the thickness, and the microstructure of ODLs was rarely explored, thus, it is worth noting and exploring. However, when the oxygen content in a furnace is high, the compound layer is easy to fall off due to the thermal expansion coefficient of the compound layer and the titanium matrix is significantly different, which is not beneficial to observe ODLs. Therefore, researchers consider that these experiments should be conducted under vacuum condition.

Based on the foregoing introduction, it is inferred that the oxygen ingress should be reduced and kept under control to eliminate the formation of a polluted layer and prevent losing too much material. Accordingly, it is of significant importance to obtain in-depth information about the effect of thermal treatment on the ODL of TC21 alloy. In this regard, the intention to explore the high-temperature oxygen uptake behaviors of TC21 alloy with a lamellar microstructure was undertaken.

## 2. Materials and Experimental Procedures

### 2.1. Results of Dilatometry and Obtained Alloy Microstructure

The chemical composition of TC21 alloy is presented in Table 1. In the present study, a dilatometer (DIL-805 A/D, Bahr, Oslo, Norway) was applied to measure β-transus temperature ($T_\beta$), and a temperature of 955 °C was obtained in this regard. Then an appropriated heat treatment process was selected accordingly. Figure 1 illustrates a typical morphology of TC21 alloy with a lamellar microstructure, which was obtained through the solution treatment at 1050 °C for 1 h, followed by a rapid water quenching (WQ).

**Table 1.** Chemical composition of TC21 alloy (wt.%).

| Elements | Al | Sn | Zr | Mo | Cr | Nb | Si | V | Ti |
|---|---|---|---|---|---|---|---|---|---|
| Percentage | 5.91 | 2.16 | 2.44 | 3.27 | 1.68 | 2.22 | 0.11 | 0.12 | Bal |

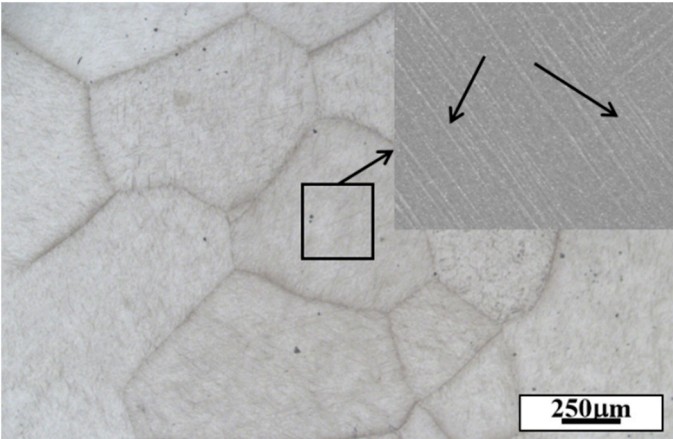

**Figure 1.** Morphology of TC21 alloy with a lamellar microstructure.

*2.2. Experimental Procedures*

In this section, samples of TC21 alloy with a lamellar microstructure were prepared. Length, width, and thickness of samples were 20 mm × 15 mm × 15 mm, respectively. Then, the surfaces of all samples were mechanically grinded. The grinding process began with 150-grit papers and proceeded with 320, 600, 800, and 2000 grits, respectively. Finally, samples were put into a vacuum quenching furnace and heat-treated at the temperature of 940, 950, 960, 980, and 1000 °C for 8 h, respectively. Then all samples were furnace cooled to room temperature. The average cooling time to 100 °C was about 4 h. It should be known that only two parameters in the furnace were able to be adjusted and controlled: temperature and time. Moreover, in this study, the vacuum degree of the furnace is always maintained at $10^{-1}$ Pa during the whole process no matter what the oxygen uptake temperature is. Therefore, temperature is the only variable when the time is the same.

Cross-sections of samples were prepared through grinding and polishing. Then samples were etched by the Kroll solution (10% HF, 30% $HNO_3$, and 70% $H_2O$) for performing detailed microstructural investigations. The microstructure of the ODL and matrix were analyzed through an optical microscope (OM) (DMI 5000 M, Leica, Wetzlar, Germany), scanning electron microscope (SEM) (SUPRA40, Zeiss, Heidenheim, Germany) and a transmission electron microscope (TEM) (G2 F20, FEI Tecnai, Hillsboro, OR, USA). Recrystallization of the microstructure in ODL and matrix were analyzed through the EBSD. In order to obtain high-quality EBSD results, samples were first electropolished and then polished with argon ion. Prior to the TEM, samples were mechanically ground to a thickness of 100 μm polished by electropolishing at $-30$ °C with a solution of $HClO_4$:$CH_3(CH_2)_3OH$:$CH_3OH$ with a ratio of 1:7:12. The cross-section hardness was tested through an ultra-micro dynamic hardness tester (211/211S, DUH, Shimadzu, Kyoto, Japan), while the test load and the corresponding holding time were set to 0.98 N and 15 s, respectively. Finally, the oxygen distribution was measured through the electron microprobe analysis (EMPA).

## 3. Results

*3.1. Microstructural Characteristics*

Figure 2 shows the cross-section morphology of TC21 alloy after oxygen uptake at different temperatures. It is observed that compared with the matrix, more equiaxed $\alpha_p$ formed in the ODL. Moreover, it also presented that there is a clear boundary between the ODL and the matrix, indicating that the formation of ODL is related to the phase transition. Meanwhile, it was found that as the temperature of oxygen uptake increased, the ODL thickness also initially increased and then decreased. The maximum thickness of ODL was 760 μm, which can be obtained for the oxygen uptake temperature of 960 °C, as shown in Table 2 and Figure 3. It is worth noting that when the oxygen uptake temperature exceeded the $T_\beta$, obvious stratifications, composed of equiaxed ODL of the outermost layer and lamellar ODL of the subsurface layer appeared. The interface between layer A and layer

B is parallel to the initial surface of the isothermal treated specimens, revealing that the diffusion direction (or the concentration gradient) of oxygen plays a dominant role in the formation of layer A. Figure 4 illustrates the SEM morphology of the ODL and the matrix of TC21 alloy with a lamellar microstructure at different temperatures. Results show more $\alpha_s$ precipitations in the ODL, indicating that the presence of oxygen enhances the precipitation of $\alpha_s$.

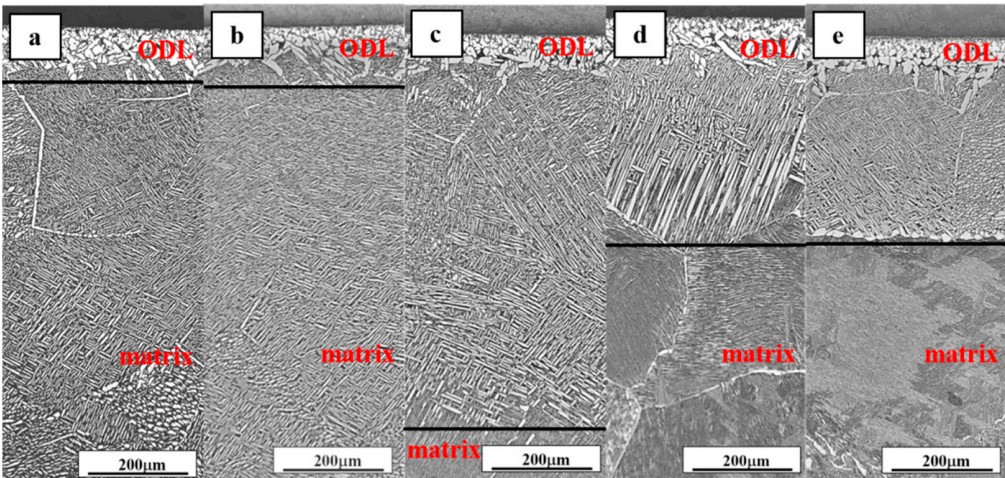

**Figure 2.** Cross-sectional morphology after oxygen ingress at different temperatures: (**a**) 940 °C, (**b**) 950 °C, (**c**) 960 °C, (**d**) 980 °C, and (**e**) 1000 °C.

**Table 2.** The oxygen diffusion layer (ODL) thickness after oxygen ingress at different temperatures.

| Temperature (°C) | 940 | 950 | 960 | 980 | 1000 |
|---|---|---|---|---|---|
| The depth of ODL (μm) | 140 | 150 | 760 | 400 | 300 |

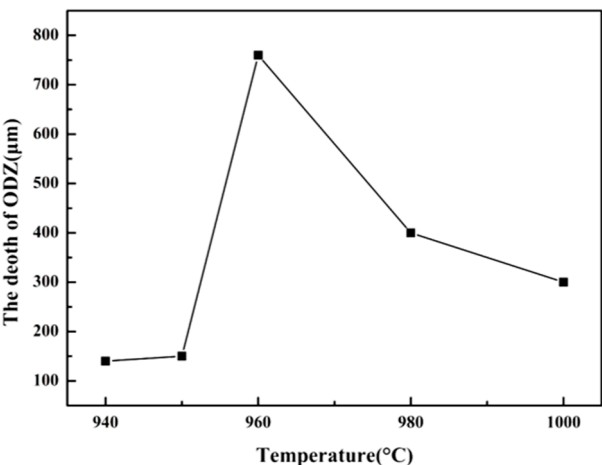

**Figure 3.** Distribution of the ODL thickness against temperature.

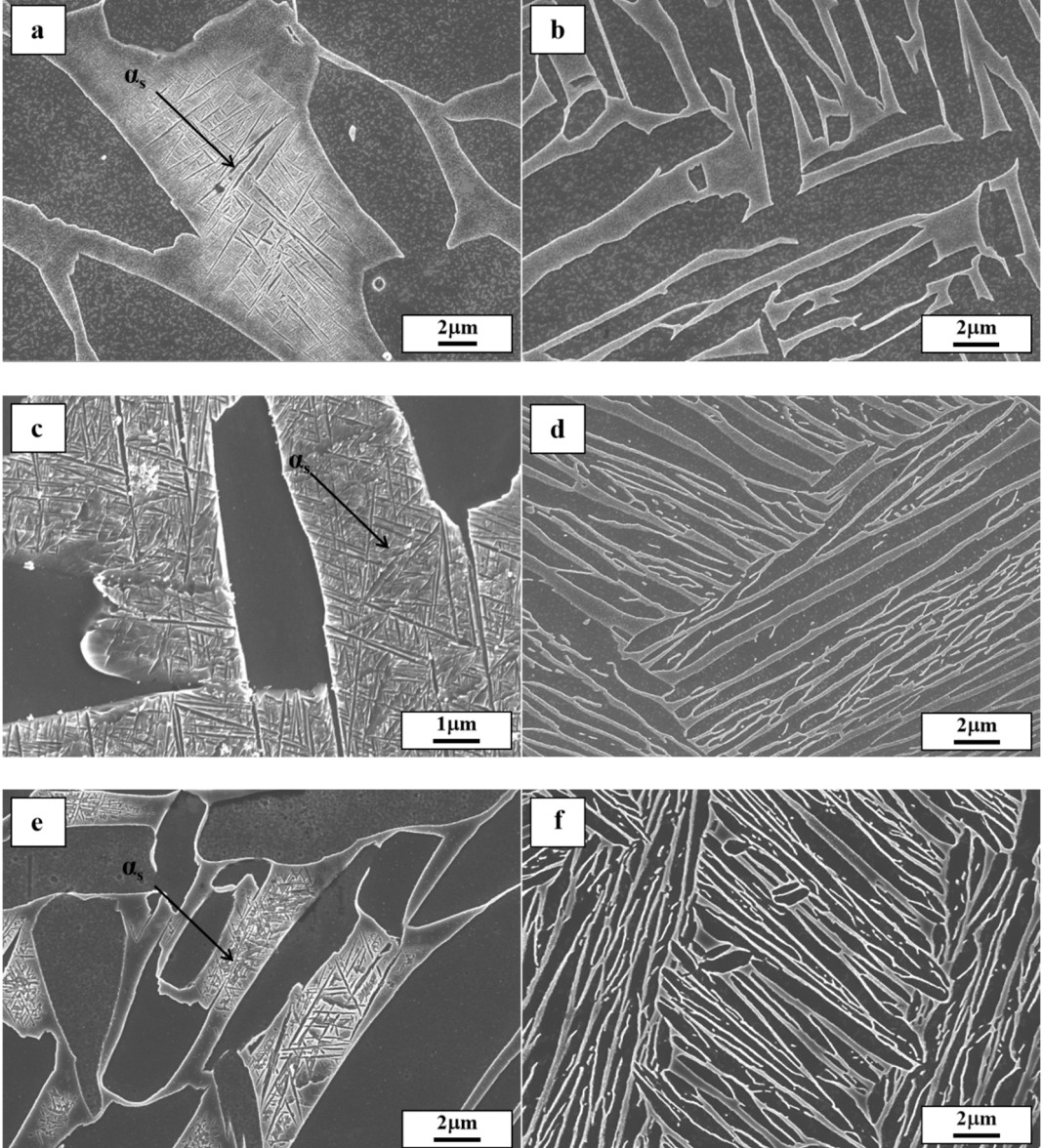

**Figure 4.** Microstructures of ODL (**a**,**c**,**e**) and matrix (**b**,**d**,**f**) after oxygen ingress at different temperatures: (**a**,**b**) 940 °C, (**c**,**d**) 960 °C, and (**e**,**f**) 1000 °C.

Figure 2 indicates that the microstructures in the ODL are significantly stratified. In order to further investigate the oxygen impact on the alloy microstructure, the ODL of the sample was analyzed for the oxygen uptake temperature of 960 °C. Figure 5 illustrates a gradient microstructure $(\alpha_p + \beta + \beta_{trans})/(\alpha_p + \beta_{trans})/(\alpha_p + \beta)$. The microstructure consists of $(\alpha_p + \beta + \beta_{trans})$ phases in Layer A, and $(\alpha_p + \beta_{trans})$ phases in Layer B, but the matrix microstructure consists of $(\alpha_p + \beta)$ phases. It is observed that all $\beta$ phases in Layer B precipitates fine $\alpha_s$, but some $\beta$ phases in Layer A do not precipitate $\alpha_s$. Compared with the matrix, a high number of nano-scale $\alpha_s$ with an average width of 30 nm were precipitated from $\beta$ phases, indicating that oxygen enhances the precipitation of $\alpha_s$. Compared with Layer B, there were more $\alpha_p$ in Layer A, but the number of $\alpha_s$ was relatively smaller. Accordingly, it can be inferred that more oxygen is required to form equiaxed $\alpha_p$, while the formation of $\alpha_s$ is relatively easy, it requires low levels of oxygen.

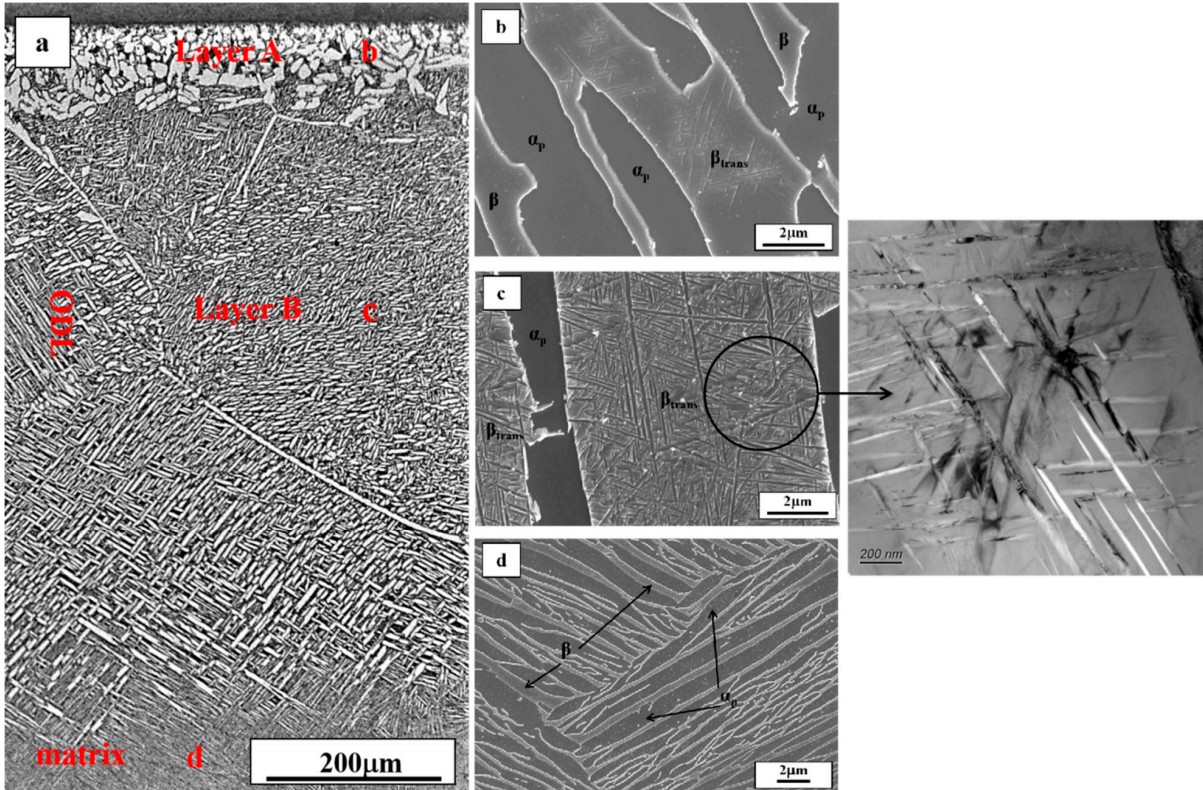

**Figure 5.** Cross-sectional morphology (**a**) of TC21 alloy after oxygen uptake at 960 °C: (**b**), (**c**) and (**d**) presents SEM morphology of Layer A, Layer B and matrix in (**a**), respectively.

Figure 6 shows that there is a grain boundary (GB) $\alpha$-$\beta$-$\alpha$-layered structure in layer A, while there is no similar phenomenon in Layer B. It is inferred that the formation of GB $\alpha$–$\beta$–$\alpha$-layered structure is closely related to the oxygen content. Figure 7 illustrates the variation of chemical composition in the layered structure. It is observed that oxygen is present in the $\alpha_p$, while the oxygen content in the $\beta$ phases is almost negligible. It is worth noting that oxygen is one of the $\alpha$-stabilizing elements, and the solid solubility of oxygen in the $\alpha$ phase is much larger than that of the $\beta$ phase.

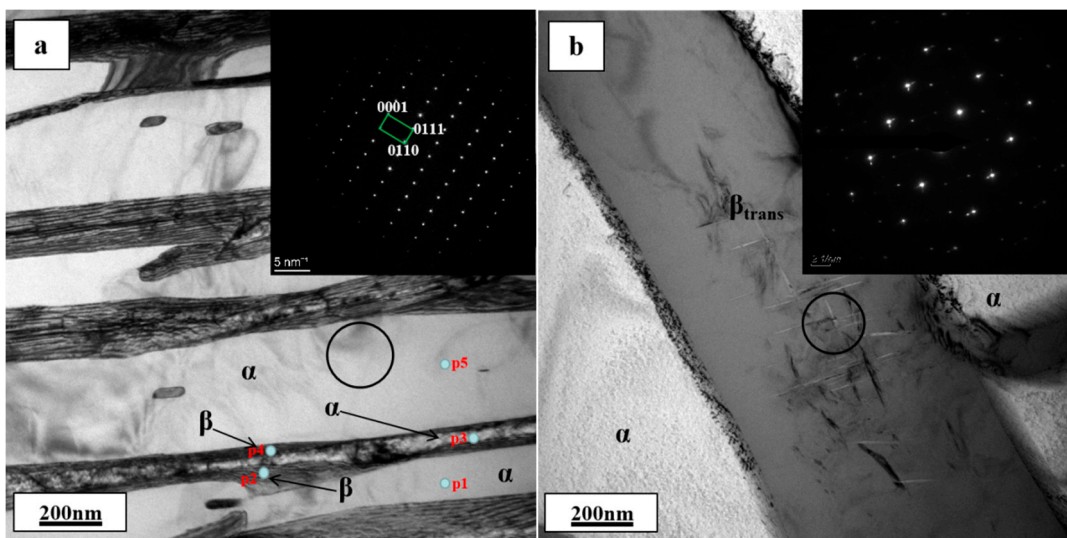

**Figure 6.** Transmission electron microscope (TEM) morphology in (**a**) layer A and (**b**) layer B after oxygen uptake at 960 °C.

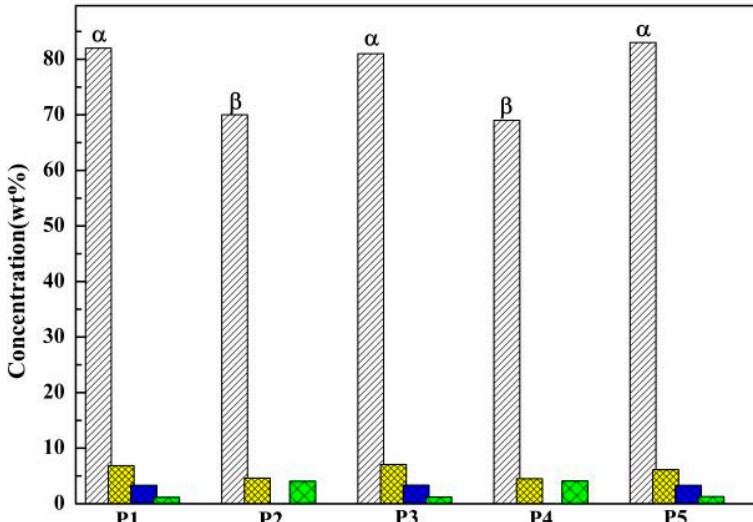

**Figure 7.** Variations of the chemical composition across the layered structure (five analyzing points, called P1, P2, P3, P4 and P5, are defined in Figure 6) showing that the α phase contains a high level of oxygen while the β phase contains a low level of oxygen (white—Ti, yellow—Al, blue—O, and green—Nb).

### 3.2. Dislocation Density and Distribution

Figure 8 presents the dislocation density and distribution of TC21 alloy after exposure to oxygen at 960 °C. It is observed that the dislocation density on the sample gradually reduces from the surface to the matrix. Moreover, it is found that the dislocation density in the ODL is much higher than that in the matrix, which mainly originates from more dislocation lines and tangles. However, the dislocation density in Layer B is relatively lower than that of Layer A. On the other hand, the dislocation distribution is also significantly different. More specifically, Figure 8a,b indicates that dislocations in Layer A are mainly in the $\alpha_p$, and there are more dislocation tangles and lines. However, Figure 8c,d shows that dislocations are mainly in the $\beta_{trans}$ and tangle $\alpha_s$ precipitations in Layer B. Meanwhile, there were only a few dislocation lines in the $\alpha_p$ so that the dislocation density of the $\alpha_p$ dramatically decreases.

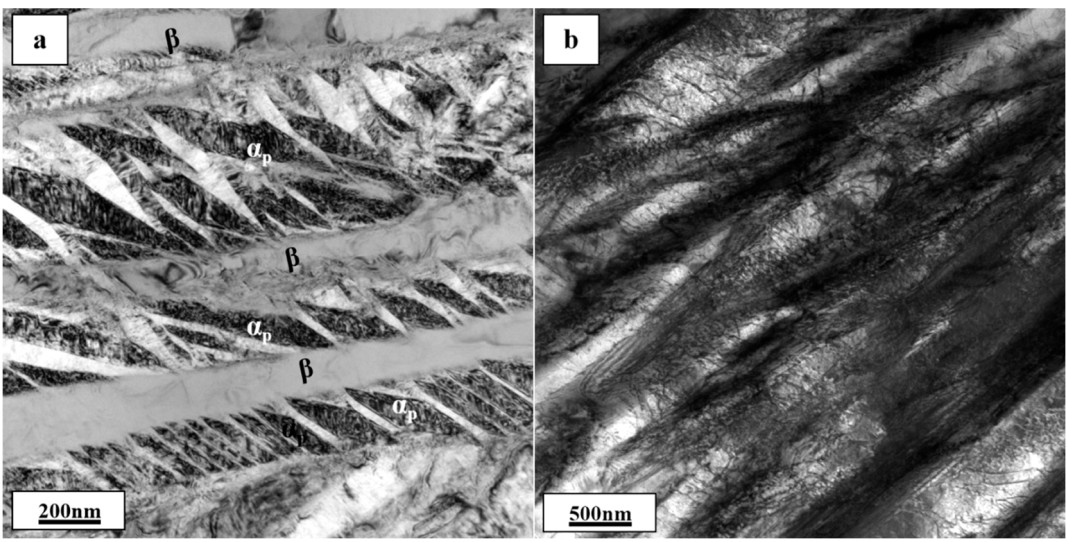

**Figure 8.** *Cont.*

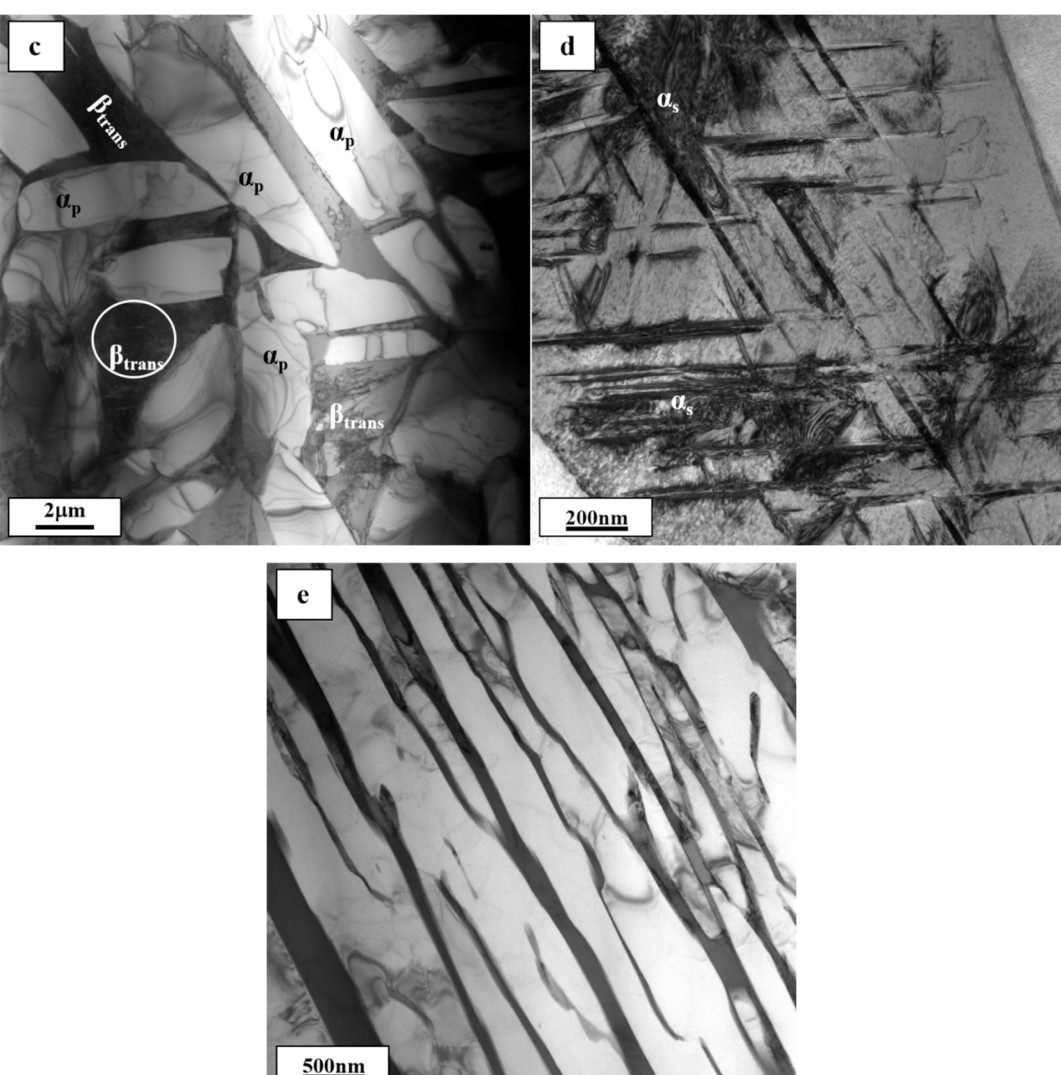

**Figure 8.** Dislocation density and distribution after oxygen uptake at 960 °C: (**a**,**b**) Layer A, (**c**,**d**) Layer B, and (**e**) matrix.

### 3.3. Mechanical Properties and Recrystallization

Figure 9 shows the variation of cross-section hardness and oxygen distribution on the sample from the surface to the matrix. Figure 9a presents ODL thicknesses of 150, 170, 740, 390, and 320 μm through hardness distribution after oxygen uptake under 940, 950, 960, 980, and 1000 °C, respectively. It is found that ODL thickness and metallographic observations have the same trend. Meanwhile, hardness values gradually decreases from the surface into the matrix, and the hardness of surface is much higher than that of the matrix. Moreover, according to Figure 9b,c, it is clearly observed that the diffusion length of oxygen exceeds 700 μm when the oxygen uptake temperature is 960 °C, but the diffusion length of oxygen is less than 200 μm at 950 °C. The hardness variation along the depth is closely correlated to the distribution of the oxygen concentration. The linear correlation between hardness and in titanium alloys has been verified [16]. Figure 10 indicates it was obvious that the volume fraction of recrystallization in the ODL is much higher than that of the matrix, which mainly originates from the decrease of substructure.

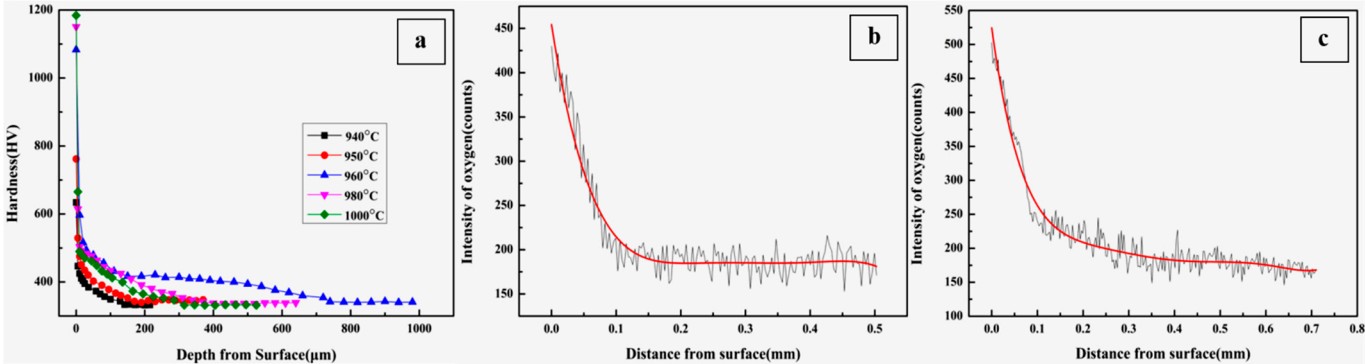

**Figure 9.** Cross-section hardness (**a**) after oxygen uptake at different temperatures and oxygen distribution, the distribution of oxygen content after oxygen uptake at (**b**) 950°C and (**c**) 960°C. Both black and red lines represent the distribution of oxygen content, the black lines are measured and given by electron microprobe analysis (EMPA), and the red lines are fitted by Origin 8.5 according to black lines.

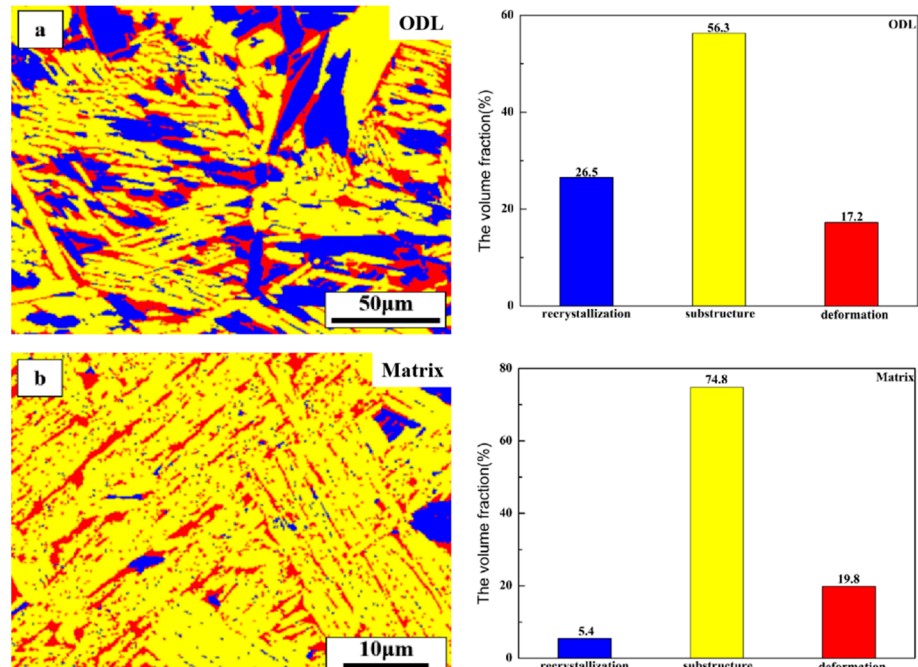

**Figure 10.** Recrystallization behavior of (**a**) ODL and (**b**) the matrix after oxygen uptake at 960 °C.

## 4. Discussion

### 4.1. The Impact of Oxygen Ingress on the Formation of $\alpha_p$, $\alpha_s$, and $\alpha$-$\beta$-$\alpha$-Layered Structures

Studies reveal that the oxygen diffusion mechanism in titanium alloys is mainly interstitial diffusion, where oxygen atoms occupy the octahedral spaces of the $\alpha$-Ti, thereby impeding the slip of dislocations and resulting in lattice distortion [17]. Meanwhile, the dislocation density in lamellar microstructure after WQ was $10^{12}$ m$^{-2}$, which is relatively high [18]. It is inferred that the alloy has high free energy and low thermodynamic stability. These factors were beneficial for the nucleation and growth of $\alpha_p$. Moreover, due to the powerful $\alpha$ stabilizing effect of oxygen, the $T_\beta$ of the localized surface region increases. For instance, this effect in TC4 alloy can be approximated by: $T_{TRANS,NEW}(°C) = 937 + 242.7 \times$ wt.% O [19]. Consequently, the oxygen-rich surface layer transforms into a continuous layer of $\alpha$ phase before the rest of the material, resulting in a characteristic brittle surface layer of continuous $\alpha$ grains with a drastic increase in the volume fraction of $\alpha$ phase within this layer [20]. However, as the distance to the surface increases, the oxygen content

rapidly decreases. Consequently, the oxygen impact on the lattice distortion and $T_\beta$ is negligible, and no $\alpha_p$ forms anymore.

In this regard, it was demonstrated that the presence of oxygen consistently enhances the precipitation of $\alpha_s$ [21]. Moreover, it was found that oxygen diffuses faster by a few orders of magnitude than the self-diffusion of Ti in both $\alpha$-Ti and $\beta$-Ti alloys [22]. Accordingly, the introduction of a fast diffuser enhances the self-diffusion of the base atoms [23]. Therefore, the improved self-diffusion kinetics of Ti atoms increases the precipitation of $\alpha_s$ [24]. Accordingly, the precipitation of acicular nano-scale $\alpha_s$ may be attributed to the presence of oxygen.

The GB $\alpha$-$\beta$-$\alpha$-layered structure can be regarded as an interface-stabilized microstructure [25], where the extra interface phase exists under the condition of the total system energy allows. However, the less stable $\beta$ phase can lead to further microstructural changes in order to reduce the total interfacial free energy. Therefore, it is normally assumed that the $\beta$ phase in the layered structure has a coherent orientation correlation with the two neighboring $\alpha$ phases, thereby generating a low interface energy. Accordingly, researchers believed that the high oxygen content in the $\alpha$–$\beta$ interfacial area could lead to an instability of the interfacial layer of the $\beta$ phase at the interface. As a result, it underwent a further transformation by transforming into the $\alpha$ phase by epitaxial growth on the existing interfacial $\alpha$ phase, thereby generating a new $\alpha$–$\beta$–$\alpha$-layered structure, while the remaining $\beta$ phase become more stable [24].

### 4.2. Influence of Temperature on the ODL Thickness

Based on the Arrhenius equation (i.e., $D = D_0 \, exp(-Q/RT)$), the higher the temperature, the greater the diffusion coefficient, and consequently the deeper the ODL thickness. However, when the oxygen uptake temperature of TC21 alloy with a lamellar microstructure exceeds $T_\beta$, as the temperature increases, the thickness of ODL gradually decreases. This is a completely abnormal phenomenon. This phenomenon can be physically interpreted as follows: firstly, as the temperature increases, the thickness of the compound layer gradually increases so that the oxygen ingress is inhibited. Figure 11 indicates that the thickness of the compound layer of TC21 alloy with a lamellar microstructure after oxygen uptake at 960 °C is 1 µm, while it is 1.8 µm at 1000 °C. Secondly, as the temperature increases, the incubation time of the compound decreases, thereby inhibiting the further ingress of oxygen [26]. Thirdly, as the temperature increases, the recrystallization decrease the density of dislocations. Consequently, oxygen diffusion depends more on the grain boundary. However, as the temperature increases, $\beta$ grains rapidly grow, thereby reducing the grain boundary. In this regard, Figure 12 shows the $\beta$ grain size against temperature. Accordingly, available channels for oxygen diffusion reduce.

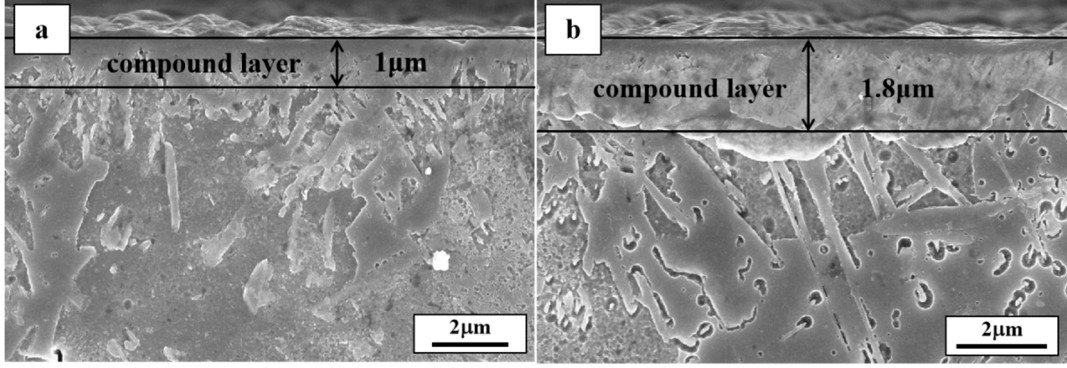

**Figure 11.** Thickness of the compound layer after oxygen uptake at (**a**) 960 °C and (**b**) 1000 °C.

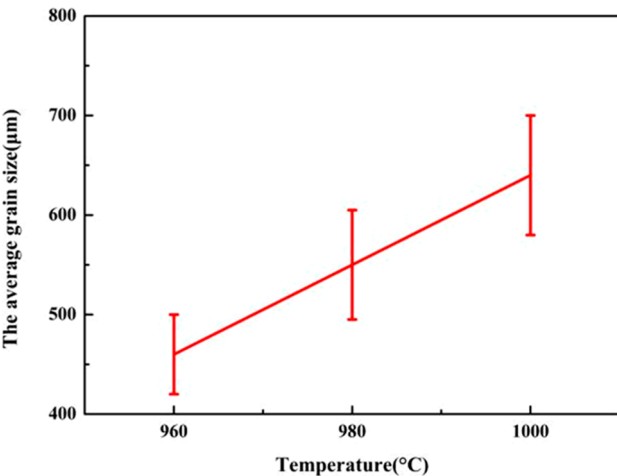

**Figure 12.** Distribution of the β grain size against temperature.

### 4.3. Influence of Oxygen on the Hardness

Studies reveal that the presence of oxygen in titanium alloys significantly increases the hardness [27]. This can be interpreted as the following: Firstly, oxygen atoms diffuse into the metal that can occupy interstitial sites. However, the dissolved oxygen may distort the lattice by increasing the c/a ratio so that the mobility of dislocations is inhibited [28]. The dissolved oxygen atoms can find more stable places around dislocations than in the rest of the lattice. This further results in "atmospheres" of atoms around dislocations [29]. When a dislocation is forced to move out of the atmosphere it causes rearrangements of the atoms in the atmosphere, which in turn distorts the lattice that increases the elastic energy. Therefore, the energy has to be added to the necessary stress so that the dislocation movement increases. Consequently, the "atmosphere" inhibits the mobility of the dislocations and makes dislocations tangle, thereby increasing the material hardness [30]. Furthermore, results indicate that this effect is more pronounced as the oxygen content increases. According to Figure 9, there were higher dislocation densities and more dislocation tangles in the ODL, indicating that oxygen atoms inhibit the dislocation motion. Therefore, oxygen ingress increases surface hardness.

### 4.4. The Anomalous Diffusion Mechanism of Oxygen at 960 °C

It was previously measured in Section 2.1 that $T_\beta$ of TC21 alloy is about 955 °C. However, the anomalous diffusion of oxygen only appeared at 960 °C, the phenomenon was not observed at 950 °C, as shown in Figure 9. Based on the result, it can be inferred that the grain boundary has little impact on the anomalous diffusion of oxygen at 960 °C. The number of grain boundaries at 960 °C was far less than that at 950 °C. On the contrary, the variations of crystal structure and diffusion rate are possible reasons for anomalous diffusion. Firstly, the microstructure completely consists of β grains for the oxygen uptake temperature of 960 °C, the diffusion rate of oxygen is faster. On the other hand, the density of the atomic arrangement in the body-centered cubic β-Ti is 0.68. This means that 68% of the β-Ti lattice capacity is occupied by titanium atoms and the remaining 32% is void. The lattice constant a of β-Ti is 0.332 nm [31], thus, the unit-cell volume of β-Ti is $a^3 = 0.0366$ nm$^3$ and the corresponding space volume Vs is $0.0366 \times 32\% = 0.0117$ nm$^3$. The lattice constant a and c of α-Ti is 0.2951 and 0.4683 nm [32], respectively. Therefore, the unit-cell volume of α-Ti is $a^3 = 0.03528$ nm$^3$ and the corresponding space volume Vs is $0.03528 \times 26\% = 0.0092$ nm$^3$ due to the density of atomic arrangement in α-Ti is 0.74. In comparison, the void volume ratio in the β-Ti is higher than that in α-Ti. In this study, researchers assumed that these larger octahedral gaps in β phases are potential channels for oxygen diffusion. Consequently, even if oxygen is not a β-stabilizing element and tends to occupy the octahedral sites in α-Ti, oxygen atoms can rapidly diffuse through these channels over great distances. Meanwhile, the change of crystal structure causes

the redistribution of alloying elements during heating and cooling, which can lead to the reformation of vacancies. Higher concentration of substitutional vacancies in the Ti matrix near $T_\beta$ facilitates the oxygen diffusion through the substitutional mechanism [33]. It is worth noting that the anomalous diffusion mechanism of oxygen near $T_\beta$ is extremely complicated and further exploration is required.

## 5. Conclusions

In the present study, microstructural variations of the TC21 alloy with a lamellar microstructure after oxygen ingress were explored and attributed to phase transformation induced by oxygen. Based on the obtained results, the following conclusions were drawn:

(1) Diffused oxygen into the substrate of TC21 alloy leads to the phase transition. The microstructural evolution indicates that oxygen not only enhances the precipitation of $\alpha_s$ but also forms more $\alpha_p$.

(2) As the temperature increases, the ODL thickness initially increases and then decreases. The maximum thickness is obtained for the oxygen uptake temperature of 960 °C.

(3) A gradient microstructure $(\alpha_p + \beta + \beta_{trans})/(\alpha_p + \beta_{trans})/(\alpha_p + \beta)$ is observed in TC21 alloy after oxygen ingress.

(4) The hardness and dislocation density of the cross-section of TC21 alloy after oxygen uptake gradually reduces from the surface to the matrix. This variation depends on the oxygen content.

**Author Contributions:** Data curation, H.S.; formal analysis, C.H.; funding acquisition, Y.L.; investigation, X.F.; methodology, S.W.; writing-original draft, S.W.; writing-review & editing, S.W. and Y.L. All authors have read and agreed to the published version of the manuscript.

**Funding:** The research documented in this work was financially supported by the central government guides local science and technology development (Grant No. [2019] 4011), the Industrial and Information Development of Guizhou Province (Grand No. [2016] 034) and the National Natural Science Foundation of China (Grant Nos. 51801037 and 52061005), and the Cultivation Project of Guizhou University (Grant No. [2019] 17).

**Institutional Review Board Statement:** Not applicable.

**Informed Consent Statement:** Informed consent was obtained from all subjects involved in the study.

**Data Availability Statement:** The data presented in this study are available on request from the corresponding author.

**Acknowledgments:** The research documented in this work was financially supported by the central government guides local science and technology development (Grant No. [2019] 4011), the Industrial and Information Development of Guizhou Province (Grand No. [2016] 034), and the National Natural Science Foundation of China (Grant Nos. 51801037 and 52061005), and the Cultivation Project of Guizhou University (Grant No. [2019] 17).

**Conflicts of Interest:** The authors declare no conflict of interest.

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
