# Peer review of "Oxygen Induced Phase Transformation in TC21 Alloy with a Lamellar Microstructure"

_metals, doi:10.3390/met11010163_

Round 1

Reviewer 1 Report

In the present article, the Authors investigated the effect of oxygen intake on phase transition in the TC21 Ti-alloy. Since the article contains quite a lot of analythical results, then there are two major concerns which, in my opinion, disqualify the article for publication in the present form. Namely:

  1. The Authors investigated oxygen intake by performing heat-treatment under vacuum conditions. Why? What was the oxygen partial pressure in the stated vacuum? The Authors did not declare any value of oxygen partial pressure. Since it is assumed that in vacuum oxygen partial pressure is quite low, then isn't the oxygen intake controlled by pO2? Are the Authors sure, that there is enough oxygen atoms available at each stage of heat-treatment to ensure the controll of oxygen intake by temperature, not pO2?
  2. The Authors showed microstructure of TC21 after heat-treatment at different temperature (Figure 3a, b, c, d, e). In all images a similar microstructure is shown, namely outer equiaxed grains below which a needle shaped microstructure is present. Why then, in figure 3 c, d, e, a needle shaped microstructure is qualified as ODL, and in figures 3a and b not? An image taken at lower magnification should be taken and the measurement of ODL zone should be re-measured. All discussion, speculations and conclusions based on this measurement are incorrect.

Moreover, during reading a following comments arrises:

  1. Page 1, line 19: Please define "ODL" abbreviation, since it is used for the first time in the text.
  2. Page 2, line 49: The Authors wrote about "non-magnetic corrosion resistance". Please define what is it non-magnetic corrosion and what is the difference between mentioned and "magnetic" corrosion.
  3. Page 2, lines 72-77: The Authors wrote, that based on introduction, the "oxygen intake should be enhanced". Are the Authors sure about such statement? Since it is negative effect, it should rather be suppressed than enhanced. The sentence is illogical.
  4. Page 2, chapter "Materials and experimental procedures": This is not a right place to show the results, like e.g. alloy microstructure or results of dilatometry..
  5. Page 3, line 87: Major comment No. 1.
  6. Page 3, Figure 3: Major comment No. 2.
  7. Page 4, Figure 4: Figure based on measurement explained in major comment No. 2.
  8. Pages 4 and 5, Figure 5: It is unknown from which regions higher magnifications are taken.
  9. Page 8, Figure 11: the Authors showed coloured microstructures of ODL and matrix including the calculated (I guess) volume fractions of phases. The graphs in the upper right corners of images revealed that the highest volume fraction was obtained for red phases. By naked eye it is vissible, that the most places in the coloured images are occupied by yellow phases, especially in the matrix. This contradicts with the graphs showing volume  fractions of phases. Please check this carefully.
  10. Page 8 line 191: please explain "WQ" abbreviation.
  11. Page 9, figure 12: The authors declared formation of oxides. Are these really oxide layers? What kind of oxide does formed? Please show analythics confirming formation of oxide scale.
  12. Chapter 4.4: whole discussion is based on incorrect measurements, like mentioned in major comment No.2.
  13. Page 11, comment No.2. Comment similar to previous one: conclusion is based on incorrect measurements of ODL from Figure 3.

Considering all mentioned comments, I reccomend to reject the article in present form and re-consider publication after re-submission incuding corrections according to the comments.

Author Response

Dear Reviewer,

Thank you for your comments concerning our manuscript entitled “Oxygen induced phase transformation in TC21 alloy with a lamellar microstructure”, which was submitted to Metals. We have answered and explained your questions carefully. Please see the attachment! 

Happy New Year to you!

Kind regards, 

Shu wang

Reviewer 2 Report

The manuscript titled "Oxygen induced phase transformation in TC21 alloy with a lamellar microstructure" is interesting and will enhance our current understanding of Ti alloys. The authors analyzed the lammellar structure of TC21 alloy and the effect of oxygen on evolution of the microstructure and hardness.  There are some minor points that must be considered before the manuscript is accepted for publication:

1) In the abstract 'ODL' is presented without giving the expanded version. I guess the authors want to talk about the oxygen diffusion layer. Please correct it.

2) There are several papers on the oxidation of Ti alloys. The authors must cite some review papers on this topic. For e.g. J. Dai, J. Zhu, C. Chen and F. Weng, Journal of Alloys and Compounds 685, 2016 (784). instead of the present citation - reference 7. 

3) In Table 1, the authors must specify if the composition is in mass% or weight%.

4) In figure 11, the text in the inset is not clear. Please modify the figure.

5) Please rephrase section 4.3 which deals with the mechanism of oxygen improved hardness. This part is not very clear.

The above mentioned points must be addressed satisfactorily before final publication. 

Author Response

Dear Reviewer,

Thank you for your comments concerning our manuscript entitled “Oxygen induced phase transformation in TC21 alloy with a lamellar microstructure”, which was submitted to Metals. We have answered and explained your questions carefully. 

Happy New Year to Editors and Reviewers!

Kind regards, 

Shu Wang

Reviewer 3 Report

Please define all the acronyms i.e. “ODl” and so on

Not sure in which industry is used and which part can be manufactured throughout  

The state of rat is very brief and not express the challenges

The novelty and your apart for the state of art is weak in the actual form

In abstract was mentioned “ electron backscatter diffraction 16 (EBSD) and transmission electron microscopy (TEM). “ but in  “Experimental procedures” no any evidence of sample preparation for these measurements !!

Please verify the entire work for typo: “Moreover, It”

You present Fig 1 then Figure 3!!! Please check it

The lamellar structure in a,b,c is very different from d,e can you explain clearly why ?

Before you suggested this alloys is formed from multiple elements however in Fig 8 you indicate only 4

Not sure what represent the small zoom in Fig 11

“Meanwhile, the dislocation density in quenching lamellar microstructure after WQ was high, about 1012 m-2” OK but we need to know in the present work the amount of dislocations …

The font size of Figure 13 is poor

Overall the discussion should be focused in what was found here not only details of what was found in literature

Author Response

Dear Reviewer,

Thank you for your comments concerning our manuscript entitled “Oxygen induced phase transformation in TC21 alloy with a lamellar microstructure”, which was submitted to Metals. We have answered and explained your questions carefully.

Happy New Year to Editor and Reviewers!

Kind regards, 

Shu Wang

Round 2

Reviewer 1 Report

The Authors adressed all questions and comments I made. Introduction of hardness measurement clarified the basis of ODL zone clasiffication. Thus I recommend to accept the paper in the present form.

Reviewer 3 Report

.